# A Theory-Based Evaluation of Nearest Neighbor Models Put Into Practice

**Hendrik Fichtenberger**[*]
TU Dortmund
Dortmund, Germany
hendrik.fichtenberger@tu-dortmund.de

**Dennis Rohde**[†]
TU Dortmund
Dortmund, Germany
dennis.rohde@cs.tu-dortmund.de

## Abstract

In the $k$-nearest neighborhood model ($k$-NN), we are given a set of points $P$, and we shall answer queries $q$ by returning the $k$ nearest neighbors of $q$ in $P$ according to some metric. This concept is crucial in many areas of data analysis and data processing, e.g., computer vision, document retrieval and machine learning. Many $k$-NN algorithms have been published and implemented, but often the relation between parameters and accuracy of the computed $k$-NN is not explicit. We study property testing of $k$-NN graphs in theory and evaluate it empirically: given a point set $P \subset \mathbb{R}^\delta$ and a directed graph $G = (P, E)$, is $G$ a $k$-NN graph, i.e., every point $p \in P$ has outgoing edges to its $k$ nearest neighbors, or is it $\epsilon$-far from being a $k$-NN graph? Here, $\epsilon$-far means that one has to change more than an $\epsilon$-fraction of the edges in order to make $G$ a $k$-NN graph. We develop a randomized algorithm with one-sided error that decides this question, i.e., a property tester for the $k$-NN property, with complexity $O(\sqrt{n}k^2/\epsilon^2)$ measured in terms of the number of vertices and edges it inspects, and we prove a lower bound of $\Omega(\sqrt{n/\epsilon k})$. We evaluate our tester empirically on the $k$-NN models computed by various algorithms and show that it can be used to detect $k$-NN models with bad accuracy in significantly less time than the building time of the $k$-NN model.

## 1 Introduction

The $k$-nearest neighborhood ($k$-NN) of a point $q$ with respect to some set of points $P$ is one of the most fundamental concepts used in data analysis tasks such as classification, regression and machine learning. In the past decades, many algorithms have been proposed in theory as well as in practice to efficiently answer $k$-NN queries [e.g., 1, 7–10, 18, 20, 23, 26, 27, 30, 36]. For example, one can construct a $k$-NN graph of a point set $P$, i.e., a directed graph $G = (P, E)$ of size $n = |P|$ such that $E$ contains an edge $(p, q)$ for every $k$-nearest neighbor $q$ of $p$ for every $p \in P$, in time $O(n \log n + kn)$ for constant dimension $\delta$ [8]. Due to restrictions on computational resources, approximations and heuristics are often used instead (see, e.g., [9, 10] and the discussion therein for details). Given the output graph $G'$ of such a randomized approximation algorithm or heuristic, one might want to check whether $G'$ resembles a $k$-NN graph before using it, e.g., in a data processing pipeline. However, the time required for exact verification might cancel out the advantages gained by using an approximation algorithm or a heuristic. On the other hand, testing whether $G'$ is at least *close* to a $k$-NN graph will suffice for many purposes. *Property testing* is a framework for the theoretical analysis of decision and verification problems that are relaxed in favor of sublinear complexity. One motivation of property testing is to fathom the theoretical foundations of efficiently assessing approximation and heuristic algorithms' outputs.

---

[*]ORCID iD: 0000-0003-3246-5323

[†]ORCID iD: 0000-0001-8984-1962

Property testing [32], and in particular property testing of graphs [21], has been studied quite extensively since its founding. A one-sided error $\epsilon$-tester for a property $\mathcal{P}$ of graphs with average degree bounded by $d$ has to accept every graph $G \in \mathcal{P}$ and it has to reject every graph $H$ that is $\epsilon$-far from $\mathcal{P}$ with probability at least $2/3$ (i.e., if graphs that are $\epsilon$-far are relevant, it has precision 1 and recall $2/3$). A graph $H$ of size $n$ is $\epsilon$-far from some property $\mathcal{P}$ if more than $\epsilon dn$ edges have to be added or removed to transform it into a graph that is in $\mathcal{P}$. A two-sided error $\epsilon$-tester may also err with probability less than $1/3$ if the graph has the property. The computational complexity of a property tester is the number of adjacency list entries it reads, denoted its *queries*. Many works in graph property testing focus on testing *plain* graphs that contain only the pure combinatorial information. However, most graphs that model real data contain some additional information that may, for example, indicate the type of an atom, the bandwidth of a data link or spatial information of an object that is represented by a vertex or an edge, respectively. In this work, we consider geometric graphs with bounded average degree. In particular, the graphs are embedded into $\mathbb{R}^\delta$, i.e., every vertex has a coordinate $x \in \mathbb{R}^\delta$. The coordinate of a vertex may be obtained by a query.

**Main Results**  Our first result is a property tester with one-sided error for the property that a given geometric graph $G$ with bounded average degree is a $k$-nearest neighborhood graph of its underlying point set (i.e., it has precision 1 and recall $2/3$ when taking $\epsilon$-far graphs as relevant).

**Theorem 1.** *Given an input graph $G = (V, E)$ of size $n = |V|$ with bounded average degree $d$, there exists a one-sided error $\epsilon$-tester that tests whether $G$ is a $k$-nearest neighbourhood graph. It has query complexity $c \cdot \sqrt{n}\, k^2 \psi_\delta / \epsilon^2$, where $\psi_\delta$ is the $\delta$-dimensional kissing number and $c > 0$ is a universal constant.*

We emphasize that it is not necessary to compute the ground truth (i.e., the $k$-NN of $P$) in order to run the property tester. Furthermore, the tester can be easily adapted for graphs $G = (P \cup Q, E)$ such that $P \cap Q = \emptyset$ and we only require that for every $q \in Q$, $E$ contains an edge $(q, p)$ for every $k$-nearest neighbor $p$ of $q$ in $P$. This is more natural when we think of $P$ as a training set and $Q$ as a test set or query domain. To complement this result, we prove a lower bound that holds even for two-sided error testers.

**Theorem 2.** *Testing whether a given input graph $G = (V, E)$ of size $n = |V|$ is a $k$-nearest neighbourhood graph with one-sided or two-sided error requires $\max(\sqrt{n/(8\epsilon k)}, k\psi_\delta/6)$ queries.*

Finally, we provide an experimental evaluation of our property tester on *approximate nearest neighbor* (ANN) indices computed by various ANN algorithms. Our results indicate that the tester requires significantly less time than the ANN algorithm to build the ANN index, most times just a $1/10$-fraction. Therefore, it can often detect badly chosen parameters of the ANN algorithm at almost no additional cost and before the ANN index is fed into the remaining data processing pipeline.

**Related Work**  We give an overview of sublinear algorithms for geometric graphs, which is the topic of research that is most relevant to our work. As mentioned above, the research on $k$-NN algorithms is very broad and diverse. See, e.g., [15, 33] for surveys. Testing whether a geometric graph that is embedded into the plane is a Euclidean minimum spanning tree has been studied by Ben-Zwi et al. [3] and Czumaj and Sohler [11]. In [3], the authors show that any non-adaptive tester has to make $\Omega(\sqrt{n})$ queries, and that any adaptive tester has query complexity $\Omega(n^{1/3})$. In [11], a one-sided eror tester with query complexity $\tilde{O}(\sqrt{n/\epsilon})$ is given. In a fashion similar to property testing, Czumaj et al. [14] estimate the weight of Euclidean Minimum Spanning Trees in $\tilde{O}(\sqrt{n} \cdot \text{poly}(\epsilon))$ time, and Czumaj and Sohler [12] approximate the weight of Metric Minimum Spanning Trees in $\tilde{O}(n \cdot \text{poly}(\epsilon))$ time for constant dimension, respectively. Hellweg et al. [22] develop a tester for Euclidean $(1 + \delta)$-spanners. Property testers for many other geometric problems can, for example, be found in [13, 29].

## 2  Preliminaries

Let $d, \delta, \epsilon, k \geq 0, d \geq k$ be fixed parameters. In this paper, we consider property testing on directed geometric graphs with bounded average degree $d$. A graph $G = (V, E)$ with an associated function $\text{coord} : V \to \mathbb{R}^d$ is a *geometric graph*, where each vertex $v$ is assigned a coordinate $\text{coord}(v)$. Given $v \in V$, we denote its degree by $\text{d}(v)$ and the set of adjacent vertices $\text{N}(v) := \{u \mid (v, u) \in E\}$. The Euclidean distance between two points $x, y$ is denoted by $\text{dist}(x, y)$. For the sake of simplicity, we

write $\mathrm{dist}(u,v) := \mathrm{dist}(\mathrm{coord}(u), \mathrm{coord}(v))$ for two vertices $u, v \in V$. When there is no ambiguity, we also refer to $\mathrm{coord}(v)$ by simply writing $v$. We denote the size of the graph $G = (V, E)$ at hand by $n = |V|$.

**Definition 1** (k-nearest neighborhood graph). A geometric graph $G = (V, E)$ is a $k$-nearest neighbourhood ($k$-NN) graph if for every $v \in V$, the $k$ points $u_1, \dots, u_k \in V$ that lie nearest to $v$ according to $\mathrm{dist}(\cdot, \cdot)$ are neighbors of $v$ in $G$, i.e., $(v, u_i) \in E$ for all $i \in [k]$ (breaking ties arbitrarily).

Let $G = (V, E)$ be a geometric graph. We say that a graph $G$ is $\epsilon$-far from a geometric graph property $\mathcal{P}$ if at least $\epsilon d n$ edges of $G$ have to be modified in order to convert it into a graph that satisfies the property $\mathcal{P}$. We assume that the graph $G$ is represented by a function $f_G : V \times [n] \to V \cup \{\star\}$, where $f_G(v, i)$ denotes the $i^{th}$ neighbor of $v$ if $v$ has at least $i$ neighbors (otherwise, $f_G(v, i) = \star$), a degree function $\mathrm{d}_G : V \to \mathbb{N}$ that outputs the degree of a vertex and a coordinate function $\mathrm{coord}_G : V \to \mathbb{R}^\delta$ that outputs the coordinates of a vertex.

**Definition 2** ($\epsilon$-tester). A one-sided (error) $\epsilon$-tester for a property $\mathcal{P}$ with query complexity $q$ is a randomized algorithm that makes $q$ queries to $f_G$, $\mathrm{d}_G$ and $\mathrm{coord}_G$ for a graph $G$. The algorithm accepts if $G$ has the property $\mathcal{P}$. If $G$ is $\epsilon$-far from $\mathcal{P}$, then it rejects with probability at least $2/3$.

The motivation to consider query complexity is that the cost of accessing the graph, e.g., through an ANN index, is costly but cannot be influenced. Therefore, one should minimize access to the graph.

**Definition 3** (witness). Let $\#\mathrm{nearer}(v, w) := |\{u \in V \mid u \neq v \wedge \mathrm{dist}(v, u) < \mathrm{dist}(v, w)\}|$ denote the number of vertices $u$ that lie nearer to $v$ than $w$. Further let $\mathrm{knn}(v) := \{u \in V \mid u \neq v \wedge \#\mathrm{nearer}(v, u) \leq k - 1\}$ denote the set of $v$'s *k-nearest neighbors*. Let $\mathrm{wit}(v) := \{u \in V \mid u \notin N(v) \wedge u \in \mathrm{knn}(v)\}$ define the subset of $\mathrm{knn}(v)$ that is not adjacent to $v$. If $\mathrm{wit}(v) \neq \emptyset$ or $\mathrm{d}(v) < k$, we call $v$ *incomplete*, and we call elements of $\mathrm{wit}(v)$ the *witnesses* of $v$.

If $G$ is $\epsilon$-far from being a $k$-nearest neighborhood graph, an $\epsilon$-fraction of its vertices are incomplete. The proof follows from common arguments in property testing (see the full version [16]).

**Lemma 4.** *If $G$ is $\epsilon$-far from being a k-nearest neighborhood graph, at least $\epsilon d n/(2k)$ vertices are incomplete.*

The main challenge for the property tester will be to find matching witnesses for a fixed set of incomplete vertices. The following result from coding theory for Euclidean codes bounds the maximum number of points $q_i$ that can have the same fixed point $p$ as nearest neighbor.

**Lemma 5.** *[35] Given a point set $P \subset \mathbb{R}^\delta$ and $p \in P$, the maximum number of points $q_i \in P$ that can have $p$ as nearest neighbour is bounded by the $\delta$-dimensional kissing number $\psi_\delta$, where $2^{0.2075\delta(1+o(1))} \leq \psi_\delta$ [34] and $\psi_\delta \leq 2^{0.401\delta(1+o(1))}$ [24] (asymptotic notation with respect to $\delta$).*

## 3  Upper Bound

The idea of the tester is as follows (see Algorithm 1). Two samples are drawn uniformly at random: $S'$, which shall contain many incomplete vertices if $G$ is $\epsilon$-far from being a $k$-nearest neighborhood graph and $T$, which shall contain at least one witness of an incomplete vertex in $S'$. For every $v \in S'$, the algorithm should query its degree, its coordinate as well as every adjacent vertex and their coordinates and calculate the distance to them. If $\mathrm{d}(k) < k$ or if one of the vertices in $T$ is a witness of $v$, the algorithm found an incomplete vertex, and hence rejects. Otherwise, it accepts.

However, we have to deal with the case that some vertices in $S'$ have non-constant degree, say, $\Omega(1/\epsilon)$, such that querying all their adjacent vertices would require too many queries. To this end, we prove that one can prune these vertices to obtain a subset $S \subseteq S'$ of *low degree* vertices that still contains many incomplete vertices with sufficient probability.

**Proof of Theorem 1**  We prove that Algorithm 1 is an $\epsilon$-tester as claimed by Theorem 1. Since Algorithm 1 does never reject a $k$-nearest neighbourhood graph, assume without loss of generality that $G = (V, E)$ is $\epsilon$-far from being a $k$-nearest neighborhood graph. Algorithm 1 only queries the neighbors of $S$, and therefore its query complexity is at most $|S| \cdot 100k/\epsilon = O(\sqrt{n}k^2/\epsilon^2)$. It remains to prove the correctness.

In the following, let $L := \{v \in V \mid \mathrm{d}(v) \leq 100k/\epsilon\}$ denote the set of all vertices in $G$ that have *low degree*, let $I$ denote the set of incomplete vertices in $L$, and let $I_S \subseteq S$ denote the set of incomplete

**Algorithm 1:** Tester for $k$-nearest neighborhood

---

**Data:** $G = (V, E)$, $d$, $k$, $\epsilon$
**Result:** *accept* or *reject*
$S' \leftarrow$ sample $\frac{100k\sqrt{n}}{\epsilon}$ vertices from $V$ u.a.r. without replacement;
$T \leftarrow$ sample $\ln(10) \cdot k \cdot \psi_\delta \cdot \sqrt{n}$ vertices from $V$ u.a.r. with replacement;
$S \leftarrow \{v \in S' \mid d(v) \leq 100k/\epsilon\}$;
**for** $v \in S, u \in T$ **do**
   **if** $(u \neq v \wedge u \in \mathrm{knn}(v) \wedge u \notin N(v)) \vee d(v) < k$ **then**
      reject;
   **end**
**end**
accept;

---

vertices in $S$. By an averaging argument, $|V \setminus L| \leq \epsilon dn/(100k)$. It follows from Lemma 4 that $L$ contains at least $\epsilon dn/(4k)$ incomplete vertices, and therefore we focus on finding incomplete vertices that have low degree. Given $u \in T$, let $W_S(u)$ be a random variable that is 1 if $u$ is a witness of an incomplete vertex $v \in I_S$ and 0 otherwise.

The proof of Theorem 1 follows from the following three claims. First, note that $S$ is a uniform sample without replacement from $L$ whose size $|S|$ is random. However, $|S|$ is sufficiently large with constant probability. This claim follows from Markov's inequality (see full version [16]).

**Claim 6.** *With probability at least* $9/10$, $|S| \geq 20\sqrt{n}/\epsilon$.

In the subsequent sections, we prove the following two claims. Given that $S$ is sufficiently large, it will contain at least $\sqrt{n}$ incomplete vertices with constant probability.

**Claim 7.** *If* $|S| \geq 20\sqrt{n}/\epsilon$, *it holds with probability at least* $9/10$ *that* $|I_S| > \sqrt{n}$.

Finally, we show that if $S$ contains at least $\sqrt{n}$ incomplete vertices, then $T$ will contain at least one witness of such an incomplete vertex with constant probability.

**Claim 8** (Lemma 11). *If* $|I_S| > \sqrt{n}$, *with probability at least* $9/10$, $Pr[\sum_{w \in T} W_S(w) > 0]$.

The correctness follows by a union bound over these three bad events.

**Analysis of the Sample S: Proof of Claim 7**

*Proof.* Since $S$ was sampled without replacement, the random variable $|I_S|$ follows the hypergeometric distribution. Let $X$ be a random variable that denotes the number of draws that are needed to obtain $\sqrt{n}$ incomplete vertices in $S$, which therefore follows the negative hypergeometric distribution. By Lemma 4, we have $\mathrm{E}[X] \leq \frac{\sqrt{n} \cdot (n+1)}{(\epsilon dn)/(2k)+1}$. By the definition of $|I_S|$ and $X$, we have $\Pr[|I_S| < \sqrt{n}] \leq \Pr[X \geq |S|]$. We apply Markov's inequality to obtain $\Pr[X \geq |S|] \leq \frac{\sqrt{n} \cdot (n+1)}{|S|(\epsilon dn)/(2k)+1}$. It follows that $|S| \in \Omega(\sqrt{n})$ ensures $|I_S| \geq \sqrt{n}$ with sufficient probability. $\square$

**Analysis of the Sample T: Proof of Claim 8**     We prove the following lower bound on the number of witnesses in $G$, which will imply a bound on $|T|$ by *k-reducing* it to the case $k = 1$.

**Proposition 9.** *Given a point set* $P \subset \mathbb{R}^\delta$, $p \in P$ *and* $k \in \mathbb{N}$, *the maximum number of points* $q_i \in P$ *that can have* $p$ *as k-nearest neighbor is bounded by* $k \cdot \psi_\delta$.

We note that this bound is tight, as shown in Lemma 12.

**Definition 10** ($k$-reducing). Let $p \in P$ be an arbitrary point. Fix $Q := \{q \in P \mid \#\mathrm{nearer}(q, p) \leq k - 1\}$. Repeat the following steps until $\forall q \in Q : \nexists q' \in Q \setminus \{q\} : \mathrm{dist}(q, q') < \mathrm{dist}(q, p)$.

   ($*$) Pick a point $q \in Q$ that lies furthest from $w$ and let $Q_q := \{q' \in Q \mid q \neq q' \wedge \mathrm{dist}(q, q') < \mathrm{dist}(q, p)\}$.

(#) Set $Q := Q \setminus Q_q$.

*Proof of Proposition 9.* We apply Definition 10 to $p$ and prove that the size of $Q$ at the beginning of the process is at most $k \cdot \psi_\delta$, which proves the claim.

At first we show that every vertex that is picked by $(*)$ stays in $Q$: Let $q_1, q_2$ be arbitrary points that are picked by $(*)$ in the process of $k$-reducing, with $q_1$ being picked in an earlier iteration than $q_2$. The latter implies $\mathrm{dist}(q_2, p) < \mathrm{dist}(q_1, p)$. Assume that $q_1 \in Q_{q_2}$ at the time $q_2$ is selected, and therefore $q_1$ is removed from $Q$. Since $q_1$ is deleted by $(\#)$, it holds that $\mathrm{dist}(q_1, p) < \mathrm{dist}(q_2, p)$, which is a contradiction as $q_1$ has been selected before $q_2$.

We continue to bound the maximum number of vertices that share their $k$-nearest neighbor: Because $p$ is the nearest point for the remaining $q \in Q$, we apply Lemma 5 and conclude that at most $\psi_\delta$ vertices are remaining in $Q$ after $k$-reducing. Since every iteration of step $(\#)$ removed at most $k-1$ points from $Q$, the cardinality of $Q$ at the beginning of the process was at most $\psi_\delta + (k-1) \cdot \psi_\delta = k \cdot \psi_\delta$. $\square$

Since at most $k \cdot \psi_\delta$ vertices can share a witness by Proposition 9, there are at least $\frac{|I_S|}{k \cdot \psi_\delta}$ distinct witnesses of vertices in $S$. We employ this bound to calculate the size of the sample $T$ such that it contains at least one witness of an incomplete vertex in $S$ with constant probability.

**Lemma 11.** *If $|S| \geq \frac{10\sqrt{n}}{\epsilon}$ and $|T| \geq \ln(10) \cdot k \cdot \psi_\delta \cdot \sqrt{n}$, then $\Pr\left[\sum_{w \in T} W_S(w) = 0\right] \leq \frac{1}{10}$.*

*Proof.* Since every vertex is sampled uniformly at random with replacement, the event that one vertex is a witness is a Bernoulli trial with probability $\Pr_{w \in V}[W_S(w) = 1] \geq \frac{|I_S|}{k \cdot \psi_\delta} \cdot \frac{1}{n} = \frac{|I_S|}{k \cdot \psi_\delta \cdot n}$. Therefore $\Pr_{w \in V}[W_S(w) = 0] \leq \left(1 - \frac{|I_S|}{k \cdot \psi_\delta \cdot n}\right)$. We have

$$
\begin{aligned}
|T| &\geq \ln(10) \cdot k \cdot \psi_\delta \cdot \sqrt{n} \\
\Rightarrow |I_S| \cdot |T| &\geq \ln(10) \cdot k \cdot \psi_\delta \cdot n & (1)
\end{aligned}
$$

$$
\Rightarrow \left(1 - \frac{|I_S|}{k \cdot \psi_\delta \cdot n}\right)^{|T|} \leq \frac{1}{10} \qquad (2)
$$

$$
\Leftrightarrow \Pr\left[\sum_{u \in T} W_S(u) = 0\right] \leq \frac{1}{10} \qquad (3)
$$

By Claim 7, Eq. (1) holds for $|S|$ as chosen in Algorithm 1. In Eq. (2) we use the fact that $1 - x \leq e^{-x}$ and in Eq. (3) we use that all events $W_S(u) = 0$ for $u \in T$ are independent Bernoulli trials. $\square$

Finally, we observe that the factor $k$ that is introduced in Proposition 9 is tight.

**Lemma 12.** *For every $\delta \geq 3, k \geq 2$, there exists a point set $P \subset \mathbb{R}^\delta$ such that there is a set of $k\psi_\delta$ points $q_i \in P$ that have the same $k$-nearest neighbor.*

*Proof.* Take a set $P = \tilde{P} \cup (0, \ldots, 0)$ of $\delta$-dimensional points, where $\tilde{P}$ consists of $\psi_\delta$ points from $\mathbb{R}^\delta$ that have $(0, \ldots, 0) \in \mathbb{R}^\delta$ as their nearest neighbor. Create a new point set $P'$ from $P$ by splitting each point $p \in P \cap \tilde{P}$ into $k$ points $p_1, \ldots, p_k$. Breaking ties arbitrarily, the 1 to $k-1$ nearest neighbors of $p_i$ are $\cup_{j \neq i}\{p_j\}$ (with distance 0), but $(0, \ldots, 0)$ is the k-nearest neighbor for all $p_i$, $i \in [k]$. Thus, $|P'| + 1 = k \cdot |\tilde{P}| + 1 = k\psi_\delta + 1$ and all points in $P'$ except the origin have $(0, \ldots, 0)$ as their $k$-nearest neighbor. $\square$

## 4 Lower Bound

We prove the first lower bound by constructing two (distributions of) graphs that are composed of multiple copies of the same building block. All graphs in one distribution are $k$-nearest neighborhood graphs, and all graphs in the other distribution are $\epsilon$-far from the property. It suffices to show that no deterministic algorithm that makes $o(\sqrt{n})$ queries can distinguish these two distributions with sufficiently high probability. Our building block is defined as follows.

**Definition 13** (line gadget)**.** Let $x \in \mathbb{R}$. A *line gadget* is a geometric, complete, directed graph $L_x = (V, E)$ of size $k+1$. The vertices $v_1, \ldots, v_{k+1} \in V$ have coordinates $x, x+1, \ldots, x+k \in \mathbb{R}$.

Note that a line gadget is a $k$-nearest neighborhood graph itself. In the following, let $k' := k+1$. The graphs in the first distribution $\mathcal{D}_1$ are composed of $n/k'$ line gadgets with sufficiently large pair-wise distances that maintain the $k$-nearest neighborhood property. The construction of the distribution of $\epsilon$-far graphs $\mathcal{D}_2$ is a bit more complicated. Basically, we want to move $\lceil \epsilon n/k' \rceil$ line gadgets to the exact position of $\lceil \epsilon n/k' \rceil$ other line gadgets such that in the resulting graph, $\lceil \epsilon n/k' \rceil$ pairs of line gadgets share the same coordinates. However, we have to make sure that the algorithm is oblivious of this relocation with sufficiently high probability. We provide a sketch of the proof here, the whole proof is contained in the full version [16].

**Lemma 14.** *Testing whether a graph is a $k$-nearest neighborhood graph with two-sided error requires $\sqrt{n/(8\epsilon k')}$ queries.*

*Proof sketch.* It is sufficient to show that for any deterministic algorithm that makes $o(\sqrt{n})$ queries, the distributions of knowledge graphs that are obtained from distributions $\mathcal{D}_1$ and $\mathcal{D}_2$, respectively, have small statistical distance.

Without loss of generality, one can assume that every query of the algorithm to a graph from $\mathcal{D}_1$ reveals a line gadget that is not in the knowledge graph yet. Then, the probability that some line gadget is revealed by the $i$-th query is uniform over all undiscovered line gadgets. Now, consider a graph from $\mathcal{D}_2$. Call all line gadgets that share their coordinates with another line gadget to be blue, and call all other line gadgets to be red. One can show that if a query does not reveal a blue line gadget such that a previous query revealed a blue line gadget with the same coordinates, then the revealed line gadget is distributed uniformly among all other undiscovered (red or blue) line gadgets.

The probability that two (blue) line gadgets with the same coordinates are revealed by the first $b = \sqrt{n/(8\epsilon k')}$ queries is upper bounded by the probability that for any pair of queries $(i,j) \in [b]^2$, query $i$ hits one of the $k'$ vertices in one of the $\lceil \epsilon n/k' \rceil$ moved line gadgets times the probability that query $j$ hits its (blue) counterpart. One can show that this probability is at most $\sum_{i,j \in [b]} \lceil \frac{\epsilon n}{k'} \rceil \cdot \frac{k'}{n} \cdot \frac{k'}{n} \leq b^2 \frac{\epsilon k'}{n} \leq 1/4$. Therefore, the total variation distance between the knowledge graph distributions is at most $1/4$. $\qquad\square$

In property testing, it is common to fix problem specific parameters such as the dimension and analyze the asymptotic behavior with respect to $n$ and $\epsilon$. However, it may be interesting that the computational complexity of a tester for $k$-nearest neighborhood graphs is at least linear in $\psi_\delta$. A sketch of the proof is provided in the full version [16].

**Lemma 15.** *Testing whether a graph of size $n$ is a $k$-nearest neighborhood graph with two-sided error requires at least $k\psi_\delta/6$ queries.*

## 5 Experiments

As discussed above, property testing aims at distinguishing perfect objects and objects that have many flaws at very small cost. Given the output of an approximate nearest neighbor (ANN) algorithm, a natural use case for a property tester is to decide whether the nearest neighbor index computed by the ANN algorithm is accurate or resolves many queries incorrectly.

Although Algorithm 1 already gives values for the sizes of $|S|$ and $|T|$, one would probably want to minimize the running time of the tester beyond worst-case analysis in practice. When used as a tool to assess an ANN index before actually putting it to work, it is also important that the tester actually reduces the total computation time compared to observing poor results at the end of the data processing pipeline (e.g., bad classification results) and starting over. Therefore, we seek to answer the following questions:

**Q1** Parameterization. What quality of ANN indices can be tested by different choices of $|S'|, |T|$?

**Q2** Performance. How does the testing time compare to the time required by the ANN algorithm?

**Setup** We implemented our property tester in C++ and integrated it into the Python framework *ANN-Benchmarks* [2, 4]. The key-idea of *ANN-Benchmarks* is to compare the quality of the indices built by ANN implementations, with respect to their running times and query-answer times. To evaluate our property tester, we chose three algorithms with the best performance observed in [2]: *KGraph* [1] and *hnsw* and *SW-graph* from the *Non-Metric Space Library* [7, 28]. All of the ANN algorithms are implemented in C / C++ and build upon nearest neighbor / proximity graphs. We computed the ground truth, i.e., a $k$-NN graph of the input data, for the Euclidean datasets *MNIST* (size 60 000, dimension 960, [25]), *Fashion-MNIST* (size 60 000, dimension 960, [31]) and *SIFT* (size 1 000 000, dimension 128, [19]) to evaluate the answers of the tester.

We ran our benchmarks on identical machines with 60 GB of free RAM guaranteed and an Intel Xeon E5-2640 v4 CPU running at 2.40 GHz (capable of running 20 concurrent threads) and measured CPU time. To minimize interference between different processes, a single instance of an ANN algorithm was run exclusively on one machine at a time.

The C++ source code of the property tester that was used for the experiments is available here [17]. The modified version of ANN-Benchmarks is available here [6].

**Q1: Parameterization of the Property Tester** We analyze how different choices for $|S'|$ and $|T|$ in Algorithm 1 affect which quality of ANN indices the tester is likely to reject. All ANN algorithms were run ten times for each choice of parameters built into ann-benchmarks (as listed in [5]) and every dataset. Then the tester was run once for each output and for every choice from $\{0.001, 0.01, 0.1\} \times \{0.05, 0.5, 5\}$ for $(c_1, c_2)$ in $|S'| = c_1 \cdot 8k\sqrt{n}$ and $|T| = c_2 \cdot k\sqrt{n}\log(10)$, with oracle access to the resulting ANN index. We chose to evaluate the tester for $k = 10$ because indices that are very close to 10-NN graphs – which is the hard case for the property tester to detect – can be computed by the ANN algorithms in reasonable time, and we support this decision by an additional experiment for $k = 50$. The ground truth, i.e., a $k$-NN graph of each dataset, and the $\epsilon$-distance (see Section 2) of each ANN index to a $k$-NN graph was computed offline.

We evaluate the recall of the property tester by distance of a tested ANN index to ground truth, where graphs that are no $k$-NN graphs are relevant (note that the tester always provides a witness when it rejects, so its precision is 1). Since the quality of an ANN index varies depending on the ANN algorithm's parameters and internal randomness, we group the computed ANN indices into buckets according to their distance to ground truth and depict the resulting recall on these classes in Fig. 1 for all datasets combined and for each dataset individually. As the oracle access that is provided to the property tester is oblivious of the underlying ANN algorithm, the figures show the combined results for all algorithms.

We observe that for distances and parameters that result in a reasonable overall recall, say, at least greater than 0.75, the property tester behaves comparable on all datasets. Since the property tester is guaranteed to have precision 1, even parameterizations with low recall on a small distance can be amplified by running the tester multiple times, possibly for different values of $c_1, c_2$. In summary, after choosing a target distance that the property tester should detect, the tested parameters seem suitable for data with dimensions up to roughly 800. For higher dimensions, it is likely advisable to apply dimensionality reduction techniques first before computing and using nearest neighbors in Euclidean space.

To get an indication of how the tester behaves for larger $k$, we conducted an additional experiment where we ran the tester on KGraph indices with $k = 50$. As one might expect, less indices are close to being a 50-NN graph than a 10-NN graph for the same sets of KGraph parameters (although the distance is normalized by $k$ and therefore it allows more errors), but the results indicate that it is also easier for the property tester to spot errors. This suggests that, at least for KGraph, errors are spread quite uniformly in the index rather than they are concentrated on some vertices.

**Q2: Performance** Consider the following scenario: an algorithm that processes data employs an ANN index. The quality of the algorithm's result (e.g., the classification rate) depends on the quality of the ANN index. However, the best parameters for the ANN algorithm are not known, and conclusions about the quality of the ANN index can only be drawn by looking at the algorithm's final result, which may be a long costly way to go. Does it pay out to run the property tester on the ANN index and recompute the index using different parameters if the tester rejects? We address this question by measuring the tester's performance. However, whether to use the tester or not

Figure 1: Recall of the property tester by $\epsilon$-distance of the ANN index to a 10-NN graph for different choices of the testers' parameters $c_1, c_2$. Distances are grouped into classes $(0, 10^{-4}], (10^{-4}, 10^{-3}], \ldots$, all of size at least 150 (lateral axis shows upper bound of the respective bucket). For example, the property tester rejected more than 95% of the ANN indices that are between 0.005-far and 0.01-far from being a 10-NN graph for $c_1 = 0.01, c_2 = 5$.

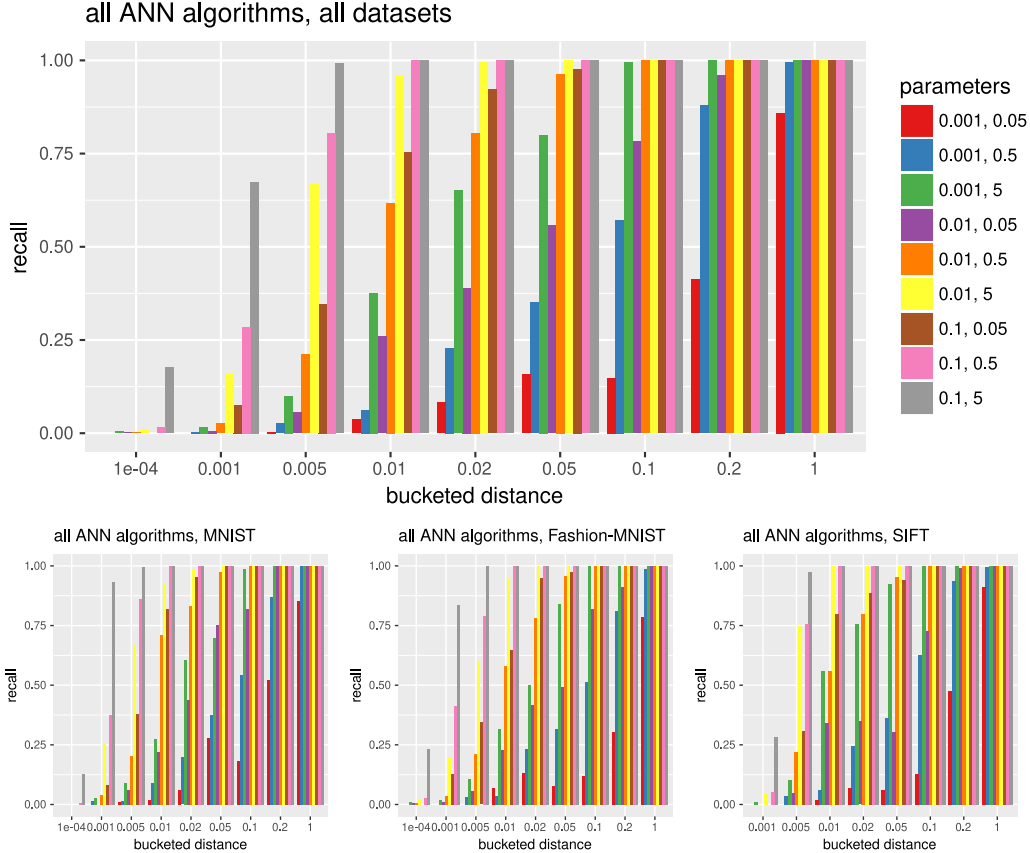

depends heavily on the cost incurred otherwise. Therefore, we compare the property tester against the minimum cost that every algorithm that uses an ANN index must invest before it can employ it or even just draw conclusions about its quality: the build time of the index. Figure 3 shows the time required by the property tester normalized (divided) by the time required to build the ANN index for each ANN algorithm and each dataset. There are two plots: one for all graphs that are between 0.005 and 0.01-close to a 10-NN graph, and one for all graphs that are between 0.01 and 0.02-close to a 10-NN graph.

In general, the running time of the property tester is always smaller than the build time for hnsw and SW-graph and at most five times the build time for KGraph. Mostly, it is even smaller than $1/10$ of the build time, and therefore running the property tester comes at almost no additional cost. For the runs of the tester on KGraph indices with $k = 50$, the testing time is also upper bounded by five times the build time and the tester time vs. build time ratio is 0.1 for $k = 10$ (restricted to MNIST and Fashion-MNIST) and $k = 50$.

## 6 Conclusion

We have studied the task of efficiently identifying NN models with low accuracy by exploring possibilities within the theoretical framework of sublinear algorithms and evaluated our approach by moving to experiments. In particular, we have proved that there is a one-sided error property tester with complexity $O(\sqrt{n}k^2/\epsilon^2)$, i.e., a sublinear (randomized) algorithm that decides whether an input graph $G$ is a $k$-NN graph or requires many edge modifications to become a $k$-NN graph (i.e.,

Figure 2: Recall of the property tester by $\epsilon$-distance of the ANN index to a $k$-NN graph for different choices of the tester's parameters and $k = \{10, 50\}$. As in Fig. 1, distances are grouped into classes.

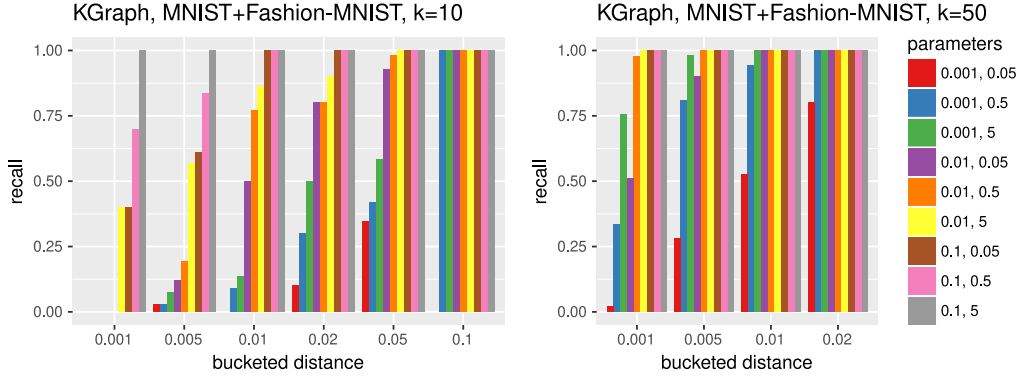

Figure 3: Performance of the property tester in terms of property tester CPU time over ANN index building CPU time for all computed ANN indices that are $(0.005, 0.01]$-far and $(0.01, 0.02]$-far from being a 10-NN graph, respectively. SW-graph computed only one graph that is 0.01-close on SIFT.

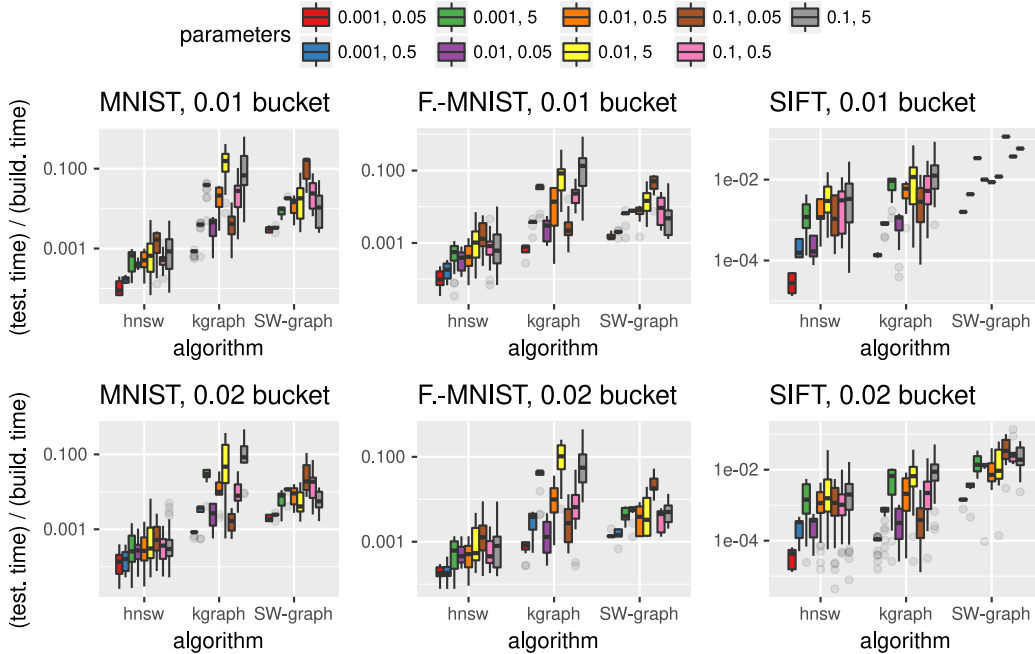

precision 1 and recall $2/3$ when taking $\epsilon$-far graphs as relevant). We also proved that even a two-sided error property tester requires complexity $\Omega(\sqrt{n/(\epsilon k)})$. Our experiments of the property tester on ANN indices computed by various algorithms indicate that testing comes at almost no additional cost, i.e., the testing time is significantly smaller than the building time of the ANN index that is tested.

From the perspective of applications, it would be desirable to analyze the tester for a more context sensitive notion of edit distance. For example, an edge to the $(k + 1)$-nearest neighbor of a point instead of an edge to its $k$-nearest neighbor might be a defect that is much less severe than an edge to the $k^2$-nearest neighbor. It would be interesting to investigate what results can be obtained under established oracle access models, which are oblivious of the graph's structure, and whether other useful models can be devised.

**Acknowledgments**

The research leading to these results has received funding from the European Research Council under the European Union's Seventh Framework Programme (FP7/2007-2013) / ERC grant agreement n° 307696. We thank the anonymous reviewers for their comments and questions, which we addressed by adding Lemma 12 and Lemma 15 most notably.

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
