[Supplementary Material]

# A Theory-Based Evaluation of Nearest Neighbor Models Put Into Practice

Hendrik Fichtenberger[*1] and Dennis Rohde[2]

[1,2]TU Dortmund, Germany
[1]hendrik.fichtenberger@tu-dortmund.de,
https://orcid.org/0000-0003-3246-5323
[2]dennis.rohde@cs.tu-dortmund.de, https://orcid.org/0000-0001-8984-1962

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

*Proof.* Assume the contrary. For every incomplete vertex $v$, delete $k$ edges such that the distance to the property does not increase and insert the missing edges from $v$ to its $k$ nearest neighbors. By the assumption, the total number of inserted or deleted edges is less than $\epsilon dn/(2k) \cdot 2k$. Therefore, $G$ is $\epsilon$-close to being a $k$-nearest neighborhood graph. □

The main challenge for the property tester will be to find matching witnesses for a fixed set of incomplete vertices. The following result from coding theory for Euclidean codes bounds the maximum number of points $q_i$ that can have the same fixed point $p$ as nearest neighbor.

**Lemma 6.** *[34] Given a point set $P \subset \mathbb{R}^\delta$ and $p \in P$, the maximum number of points $q_i \in P$ that can have $p$ as nearest neighbour is bounded by the $\delta$-dimensional kissing number $\psi_\delta$, where $2^{0.2075\delta(1+o(1))} \leq \psi_\delta$ [33] and $\psi_\delta \leq 2^{0.401\delta(1+o(1))}$ [23] (asymptotic notation with respect to $\delta$).*

# 3 Upper Bound

The idea of the tester is as follows (see Algorithm 1). Two samples are drawn uniformly at random: $S'$, which shall contain many incomplete vertices if $G$ is $\epsilon$-far from being a $k$-nearest neighborhood graph and $T$, which shall contain at least one witness of an incomplete vertex in $S'$. For every $v \in S'$, the algorithm should query its degree, its coordinate as well as every adjacent vertex and their coordinates and calculate the distance to them. If $\mathrm{d}(k) < k$ or if one of the vertices in $T$ is a witness of $v$, the algorithm found an incomplete vertex, and hence rejects. Otherwise, it accepts.

However, we have to deal with the case that some vertices in $S'$ have non-constant degree, say, $\Omega(1/\epsilon)$, such that querying all their adjacent vertices would require too many queries. To this end, we prove that one can prune these vertices to obtain a subset $S \subseteq S'$ of *low degree* vertices that still contains many incomplete vertices with sufficient probability.

---

**Algorithm 1:** Tester for $k$-nearest neighborhood

**Data:** $G = (V, E)$, $d$, $k$, $\epsilon$

**Result:** *accept* or *reject*

$S' \leftarrow$ sample $\frac{100k\sqrt{n}}{\epsilon}$ vertices from $V$ u.a.r. without replacement;

$T \leftarrow$ sample $\ln(10) \cdot k \cdot \psi_\delta \cdot \sqrt{n}$ vertices from $V$ u.a.r. with replacement;

$S \leftarrow \{v \in S' \mid \mathrm{d}(v) \leq 100k/\epsilon\}$;

**for** $v \in S, u \in T$ **do**

    **if** $(u \neq v \wedge u \in \mathrm{knn}(v) \wedge u \notin N(v)) \vee \mathrm{d}(v) < k$ **then**

        reject;

    **end**

**end**

accept;

---

**Proof of Theorem 1**   We prove that Algorithm 1 is an $\epsilon$-tester as claimed by Theorem 1. Since Algorithm 1 does never reject a $k$-nearest neighbourhood graph, assume without loss of generality that $G = (V, E)$ is $\epsilon$-far from being a $k$-nearest neighborhood graph. Algorithm 1 only queries the neighbors of $S$, and therefore its query complexity is at most $|S| \cdot 100k/\epsilon = O(\sqrt{n}k^2/\epsilon^2)$. It remains to prove the correctness.

In the following, let $L := \{v \in V \mid \mathrm{d}(v) \le 100k/\epsilon\}$ denote the set of all vertices in $G$ that have *low degree*, let $I$ denote the set of incomplete vertices in $L$, and let $I_S \subseteq S$ denote the set of incomplete vertices in $S$. By an averaging argument, $|V \backslash L| \le \epsilon dn/(100k)$. It follows from Lemma 5 that $L$ contains at least $\epsilon dn/(4k)$ incomplete vertices, and therefore we focus on finding incomplete vertices that have low degree. The following random variable identifies witnesses of vertices incomplete vertices in $S \subset L$.

**Definition 7.** Given $u \in T$, let $W_S(u)$ be a random variable that is 1 if $u$ is a witness of an incomplete vertex $v \in I_S$ and 0 otherwise.

The proof of Theorem 1 follows from the following three claims. First, note that $S$ is a uniform sample without replacement from $L$ whose size $|S|$ is random. However, $|S|$ is sufficiently large with constant probability.

**Claim 8.** *With probability at least 9/10, $|S| \ge 20\sqrt{n}/\epsilon$.*

*Proof.* The expected cardinality of $S' \backslash S$ is $d\sqrt{n}$. Therefore, the probability that $|S|$ is less than $20\sqrt{n}/\epsilon$ is at most $1/10$ by Markov's inequality.                                    $\square$

In the subsequent sections, we prove the following two claims. Given that $S$ is sufficiently large, it will contain at least $\sqrt{n}$ incomplete vertices with constant probability.

**Claim 9.** *[Lemma 11] If $|S| \ge 20\sqrt{n}/\epsilon$, it holds with probability at least 9/10 that $|I_S| > \sqrt{n}$.*

Finally, we show that if $S$ contains at least $\sqrt{n}$ incomplete vertices, then $T$ will contain at least one witness of such an incomplete vertex with constant probability.

**Claim 10** (Lemma 14)**.** *If $|I_S| > \sqrt{n}$, with probability at least 9/10, $Pr[\sum_{w \in T} W_S(w) > 0]$.*

The correctness follows by a union bound over these three bad events.

**Analysis of the Sample S: Proof of Claim 9**   We bound the cardinality of $S$ such that $S$ contains at least $\sqrt{n}$ incomplete vertices.

**Lemma 11.** *If $|S| \ge \frac{10\sqrt{n}}{\epsilon}$, then $|I_S| \ge \sqrt{n}$ with probability at least 9/10.*

*Proof.* Since $S$ was sampled without replacement, the random variable $|I_S|$ follows the hypergeometric distribution. Let $X$ be a random variable that denotes the number of draws that are needed to obtain $\sqrt{n}$ incomplete vertices in $S$, which therefore follows the negative hypergeometric distribution. By Lemma 5, we have $\mathrm{E}[X] \le \frac{\sqrt{n} \cdot (n+1)}{(\epsilon dn)/(2k)+1}$. By the definition of $|I_S|$ and $X$, we have $\Pr[|I_S| < \sqrt{n}] \le \Pr[X \ge |S|]$. We apply Markov's inequality to obtain $\Pr[X \ge |S|] \le \frac{\sqrt{n} \cdot (n+1)}{|S|(\epsilon dn)/(2k)+1}$. It follows that $|S| \in \Omega(\sqrt{n})$ ensures $|I_S| \ge \sqrt{n}$ with sufficient

probability.

$$|S| \geq \frac{20\sqrt{n}}{\epsilon}$$

$$\Leftrightarrow |S| \geq \frac{20\sqrt{n}(dn/(2k)+1)}{\epsilon(dn/(2k)+1)}$$

$$\Rightarrow |S| \geq \frac{10\sqrt{n}n + 10\sqrt{n}}{(\epsilon dn)/(2k)+1}$$

$$\Leftrightarrow \frac{1}{10} \geq \frac{\frac{\sqrt{n}(n+1)}{(\epsilon dn)/(2k)+1}}{|S|}$$

$$\Rightarrow \Pr[X \geq |S|] \leq \frac{1}{10}$$

$\square$

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

# 4 Lower Bound

We prove the first lower bound by constructing two (distributions of) graphs that are composed of multiple copies of the same building block. All graphs in one distribution are $k$-nearest neighborhood graphs, and all graphs in the other distribution are $\epsilon$-far from the property. It suffices to show that no deterministic algorithm that makes $o(\sqrt{n})$ queries can distinguish these two distributions with sufficiently high probability. Our building block is defined as follows.

**Definition 16** (line gadget)**.** Let $x \in \mathbb{R}$. A *line gadget* is a geometric, complete, directed graph $L_x = (V, E)$ of size $k + 1$. The vertices $v_1, \ldots, v_{k+1} \in V$ have coordinates $x, x + 1, \ldots, x + k \in \mathbb{R}$.

Note that a line gadget is a $k$-nearest neighborhood graph itself. In the following, let $k' := k + 1$. The graphs in the first distribution $\mathcal{D}_1$ are composed of $n/k'$ line gadgets with sufficiently large pair-wise distances that maintain the $k$-nearest neighborhood property. The construction of the distribution of $\epsilon$-far graphs $\mathcal{D}_2$ is a bit more complicated. Basically, we want to move $\lceil \epsilon n/k' \rceil$ line gadgets to the exact position of $\lceil \epsilon n/k' \rceil$ other line gadgets such that in the resulting graph, $\lceil \epsilon n/k' \rceil$ pairs of line gadgets share the same coordinates. However, we have to make sure that the algorithm is oblivious of this relocation with sufficiently high probability.

**Lemma 17.** *Testing whether a graph is a k-nearest neighborhood graph with two-sided error requires $\sqrt{n/(8\epsilon k')}$ queries.*

*Proof.* For the sake of simplicity, let $n$ be a multiple of $k'$. Let $G = (V, E)$ be a graph that is composed of $n/k'$ line gadgets $L_{3k'i}$ for $i \in [n/k']$, and let $\mathcal{D}_1$ be the uniform distribution over all vertex labellings of $G$. All graphs in $\mathcal{D}_1$ are $k$-nearest neighborhood graphs. For every graph $G \in \mathcal{D}_1$, we define a random graph $G'$ as follows. Let $S(G) = (s_i)_{i \in [2\lceil \epsilon n/k' \rceil]}$ be a sequence of random numbers, drawn without replacement from the uniform distribution over $[n]$. For every $i \in \lceil \epsilon n/k' \rceil$, move the line gadget $L_{3k's_i}$ to the coordinates of $L_{3k's_{i+\lceil \epsilon n/k' \rceil}}$ such that $G'$ contains no $L_{3k's_i}$ but two $L_{3k's_{i+\lceil \epsilon n/k' \rceil}}$ afterwards. Note that $G'$ is $\epsilon$-far from being a $k$-nearest neighborhood graph. Let $\mathcal{D}_2$ be the uniform distribution over $\bigcup_{G \in \mathcal{D}_1} G'$.

We may assume that if the tester queries for (a neighbor of) a vertex $v$, then the oracle returns the whole line gadget that $v$ belongs to. This is only beneficial for the query complexity of the algorithm. We consider the knowledge graph of the algorithm, which is defined as the subgraph of the input graph that consists of the vertices, edges and non-edges that are revealed by the answers to the queries. Without loss of generality, we may assume that the algorithm only asks queries whose answers cannot be deduced from the knowledge graph. To prove the theorem, it is sufficient to show that for any deterministic algorithm that makes $o(\sqrt{n})$ queries, the distributions of knowledge graphs that are obtained from distributions $\mathcal{D}_1$ and $\mathcal{D}_2$, respectively, have small statistical distance.

First, note that every query of the algorithm to a graph from $\mathcal{D}_1$ reveals a line gadget that is not in the knowledge graph yet. Let $R$ be the set of revealed vertices after the $i$-th query $q_i$. Then, the probability that $q_{i+1}$ reveals a line gadget $L_{3k'j}$ is $k'/(n - k'i)$ if $L_{3k'j} \notin R$ and 0 otherwise.

We turn to $\mathcal{D}_2$ now. Consider the $i$-th query $q_i$. We claim that conditioned on the event that $q_1, \ldots, q_{i+1}$ do not reveal two instances of a line gadget that is contained twice in the graph, then the line gadget that is revealed by $q_{i+1}$ is distributed uniformly. Given $G \in \mathcal{D}_2$, let $S_j(G)$ be the $j$-th half of $S(G)$ for $j \in \{1, 2\}$ such that $S(G)$ is the concatenation of $S_1(G)$ and $S_2(G)$, and let $S_2'(G)$ denote the subset of $S_2(G)$ that has already been revealed. Let $\mathcal{D}_2'$ be the restriction of $\mathcal{D}_2$ to the graphs that are compatible with the current knowledge graph, and the let $\ell$ be the support size of $\mathcal{D}_2'$. We denote the event that a line gadget $L_{3k'j}$ is revealed by $E_j$. Fix some arbitrary $j$. We have

$$\Pr[E_j] = \sum_{G \in \mathcal{D}_2'} \frac{1}{\ell} \cdot \Pr[E_j \mid j \notin S_2'(G)]$$

$$= \frac{1}{\ell} \left[ \sum_{\substack{G \in \mathcal{D}_2' \\ j \in S_1(G)}} 0 + \sum_{\substack{G \in \mathcal{D}_2' \\ j \in S_2(G)}} \frac{2k'}{n - k'i - k'|S_2'(G)|} + \sum_{\substack{G \in \mathcal{D}_2' \\ j \notin S(G)}} \frac{k'}{n - k'i - k'|S_2'(G)|} \right] = \frac{k'}{n - k'i - k'|S_2'(G)|}$$

Therefore, the knowledge graph distribution of $\mathcal{D}_2$ is exactly the same as the distribution of $\mathcal{D}_1$, i.e., uniform over all graphs that are compatible with current knowledge graph, as long as the algorithm does not reveal two line gadgets with the same coordinates. The probability that two line gadgets with the same coordinates are revealed by the first $b = \sqrt{n/(8\epsilon k')}$ queries is upper bounded by the probability that for any pair of queries $(i, j) \in [b]^2$, query $i$ hits one of the $k'$ vertices in one of the $\lceil \epsilon n/k' \rceil$ line gadgets $L_{3k's_{\ell + \lceil \epsilon n/k' \rceil}}$ such that $\ell \in \lceil \epsilon n/k' \rceil$, times the probability that query $j$ hits the same gadget. In particular, by the union bound this probability is at most $\sum_{i,j \in [b]} \lceil \frac{\epsilon n}{k'} \rceil \cdot \frac{k'}{n} \cdot \frac{k'}{n} \le b^2 \frac{\epsilon k'}{n} \le 1/4$. Therefore, the total variation distance between the knowledge graph distributions is at most $1/4$. □

In property testing, it is common to fix problem specific parameters such as the dimension and analyze the asymptotic behavior with respect to $n$ and $\epsilon$. However, it may be interesting that the computational complexity of a tester for $k$-nearest neighborhood graphs is at least linear in $\psi_\delta$.

**Lemma 18.** *Testing whether a graph of size $n$ is a $k$-nearest neighborhood graph with two-sided error requires at least $k\psi_\delta/6$ queries.*

*Proof sketch.* Let $P'(q)$ be a point set as constructed in the proof of Lemma 15 with the common $k$-nearest neighbor located at $q$. Without loss of generality, assume that $n = (c+1) \cdot |P'|$ for some $c \in \mathbb{N}$. Let $q_i = (i, 0, \ldots, 0) \in \mathbb{R}^\delta$ for $i \in [c]$. By scaling $P'$ accordingly, we can construct a set $R = P'(q_1) \cup \ldots \cup P'(q_c)$ such that for every $i \in [c]$ and every $p \in R \cap P'(q_i) \setminus \{q_i\}$, $q_i \in Q$ is the $k$-nearest neighbor of $p$. Let $G'$ be the $k$-nearest neighborhood graph of $R$.

We construct two graphs, $G$ and $H$, and prove that one requires at least $k\psi_\delta/6$ queries to distinguish between uniform distributions over all labellings of $G$ and $H$, respectively. For $i \in [\lceil 2\epsilon c \rceil]$ and some sufficiently small $\delta$, we move $q_i$ to $q_i' = (i - \delta, 0, \ldots)$ and insert a new point $q_i'' = (i + \delta, 0, \ldots, 0)$. Now, at least half of the vertices from $P'(q_i)$ have one of these points as $k$-nearest neighbor. Without loss of generality, let $q_i''$ be this point. We obtain $G$ by applying this modification to $G'$. We obtain $H$ from $G$ by moving $q_i''$ for every $i \in [\lceil 2\epsilon c \rceil]$ to $(-1, 0, \ldots, 0)$ and recalculating the $k$-nearest neighbors for all these points. It follows that $H$ is a $k$-nearest neighborhood graph, while $G$ is $\epsilon$-far from being a $k$-nearest neighborhood graph.

By the union bound, the probability to sample a point from $Q'' = \cup_{i \in [\lceil 2\epsilon c \rceil]} \{q_i''\}$ is upper bounded by $\frac{3\epsilon c}{n} \le \frac{3n}{n|P'|} \le \frac{3}{k\psi_\delta}$. Applying the union bound once again, it follows that the first $k\psi_\delta/6$ queries to $G$ or $H$ will not contain any point from $Q''$ with probability greater than $1/3$. $\square$