[Reviews · NeurIPS 2018]

Reviewer 1



SUMMARY: The paper studies the problem of testing whether a graph is epsilon-far from a kNN graph, where epsilon-far means that at least epsilon-fraction of the edges need to be changed in order to make the graph a kNN graph. The paper presents an algorithm with an upper bound of O(\sqrt{n}*k^2/\epsilon^2) number of edge/vertex queries and a lower bound of \Omega(\sqrt{n}). COMMENTS: 1. Ln 175. I guess "\omega" should be "p" 2. The result of Proposition 12 is interesting which bounds the number of points that can be in the kNN set of a particular point. The bound is k times the known bound for 1-NN. I wonder if this could be tightened somehow. Also, are there lower bounds for the 1-NN result? 3. Is there a reason why the \epsilon is hidden from the lower bound? 4. It looks like these result don't explicitly show the dependence on the dimension of the space. I guess similar comments about the lower bound: could we get some lower bounds that depend on the dimension of the space? Currently the lower bound is in 1-dimension. I guess we will have some factor that depends exponentially in the dimension of the space. 5. In practice, some mistakes are more costly than others. For example, deleting the edge to the k-th nearest neighbor and adding in an edge to the (k+1)-th nearest neighbor is not as bad as deleting the edge to the k-th nearest neighbor and adding an edge to the farthest point. Can the analysis be extended to take this into account and e.g. have some weighted cost? OVERALL: Testing whether a graph is approximately a kNN graph is an interesting problem with implications to approximate nearest neighbor (ANN) procedures. The paper provides an algorithm which tests whether such an ANN index actually approximates the exact kNN method which has reasonable computational cost which could have applications in cheaply testing whether an ANN index is acceptable for usage. The algorithm and theoretical results presented are to my knowledge novel. These results come with some experimental results on a number of datasets which suggest that this may have relevence in practice.

Reviewer 2



It is important to note that the dimension \delta is constant. applications using an infinite-dimensional feature space (e.g. when the distnace function emerges from an exponential or gaussian kernel etc.) typically have \delta close to n and aren't covered by the presented theory. The presented theory is interesting, the practical value is not fully clear (why do we want to test whether a graph is a k-nn graph?). The derivation (and full version of the derivation) may benefit from more precision and detail. Minor comments: * I guess definition 3 (knn(v)) has problems when the distance function evaluates to the same value for several neighbors of v. A comment on tie-breaking may be needed. * end of definition 3: if d(v)

Reviewer 3



This paper concerns the problem of finding the k-NN graph of a point set, i.e. a directed graph where there is an edge from a to b if b is among the k points nearest (according to some metric) to a, important in various data analysis tasks. The contribution consists in a randomized algorithm that given a graph G in sublinear time (square root n times k squared times 1/epsilon squared) will check if G is close to being a k-NN graph, i.e. what is called a property tester for k-NN graphs. A lower bound on the runtime of such an algorithm is also included (square root of n), that shows the given algorithm not far from optimal, and finally an experimental evaluation is performed that shows positive results. The given algorithm is quite simple, however the analysis is quite technical. The experimental evaluation shows that the time spent for the property testing is about 1/10 th of the time spent to find G, which seems like a good trade-off. This paper may indeed be of interest to parts of the NIPS audience.